# An Exact Hypergraph Matching algorithm for posture identification in embryonic *C. elegans*

**Andrew Lauziere**[1,2]*, **Ryan Christensen**[2], **Hari Shroff**[2¤], **Radu Balan**[1,3]

**1** Department of Mathematics, University of Maryland, College Park, MD, United States of America, **2** Laboratory of High Resolution Optical Imaging, National Institute of Biomedical Imaging and Bioengineering, National Institutes of Health, Bethesda, MD, United States of America, **3** Center for Scientific Computation and Mathematical Modeling (CSCAMM), University of Maryland, College Park, MD, United States of America

¤ Current address: Janelia Research Campus, Howard Hughes Medical Institute, Ashburn, Virginia, United States of America

* lauziere@umd.edu

**Data Availability Statement:** The code and data are available at https://github.com/lauziere/EHGM.

**Funding:** Andrew Lauziere's contribution to this research was supported in part by NSF award DGE-1632976. Radu Balan's contribution to this

## Abstract

The nematode *Caenorhabditis elegans* (*C. elegans*) is a model organism used frequently in developmental biology and neurobiology [White, (1986), Sulston, (1983), Chisholm, (2016) and Rapti, (2020)]. The *C. elegans* embryo can be used for cell tracking studies to understand how cell movement drives the development of specific embryonic tissues. Analyses in late-stage development are complicated by bouts of rapid twitching motions which invalidate traditional cell tracking approaches. However, the embryo possesses a small set of cells which may be identified, thereby defining the coiled embryo's posture [Christensen, 2015]. The posture serves as a frame of reference, facilitating cell tracking even in the presence of twitching. Posture identification is nevertheless challenging due to the complete repositioning of the embryo between sampled images. Current approaches to posture identification rely on time-consuming manual efforts by trained users which limits the efficiency of subsequent cell tracking. Here, we cast posture identification as a point-set matching task in which coordinates of seam cell nuclei are identified to jointly recover the posture. Most point-set matching methods comprise coherent point transformations that use low order objective functions [Zhou, (2016) and Zhang, (2019)]. Hypergraphs, an extension of traditional graphs, allow more intricate modeling of relationships between objects, yet existing hypergraphical point-set matching methods are limited to heuristic algorithms which do not easily scale to handle higher degree hypergraphs [Duchenne, (2010), Chertok, (2010) and Lee, (2011)]. Our algorithm, *Exact Hypergraph Matching* (*EHGM*), adapts the classical branch-and-bound paradigm to dynamically identify a globally optimal correspondence between point-sets under an arbitrarily intricate hypergraphical model. *EHGM* with hypergraphical models inspired by *C. elegans* embryo shape identified posture more accurately (56%) than established point-set matching methods (27%), correctly identifying twice as many sampled postures as a leading graphical approach. Posterior region seeding empowered *EHGM* to correctly identify 78% of postures while reducing runtime, demonstrating the efficacy of the method on a cutting-edge problem in developmental biology.

research was supported in part by NSF under grants DMS-1816608 and DMS-2108900, and by Simons Foundation under Simons Fellows in Mathematics program. The funders had no role in study design, data collection and analysis, decision to publish, or preparation of the manuscript.

**Competing interests:** The authors have declared that no competing interests exist.

## Introduction

Point-set matching describes the task of finding an optimal alignment between two sets of points. The problem appears in computer vision applications such as point-set registration [1], object recognition [2], and multiple object tracking [3]. Often the point-sets are modeled via *graphs*, abstract mathematical objects in which points are represented as vertices and edges define relationships between pairs of vertices.

User defined attributes characterize the vertices and edges, such as coordinate positions or shape descriptions and lengths of chords connecting vertices, respectively. Specified attributes give insight to observable relationships between vertices and allow for structural analyses of graphs. Graph matching is the optimization problem defined by the search for a correspondence of vertices between a pair of attributed graphs. The optimization problem uses binary variables $x_{ij}$ to specify a matching between vertex $i$ in the first graph to vertex $j$ of the second. The graph matching domain, $\mathcal{X}$, consists of assignment matrices of size $n_1 \times n_2$, for matching graphs of size $n_1$ and $n_2$:

$$\mathcal{X} = \{X \in \{0,1\}^{n_1 \times n_2} : \forall j, \sum_{i=1}^{n_1} x_{ij} \leq 1, \forall i \sum_{j=1}^{n_2} x_{ij} = 1\}. \tag{1}$$

The assignment matrix space, $\mathcal{X}$ (Eq 1), comprises assignment matrices which each describe a one-to-one alignment between nodes of the two graphs. The specification of the graph matching optimization objective function allows for joint assignment costs: i.e., how the assignment of a pair of vertex-to-vertex assignments changes the quality of the match. Let $\mathbf{C}$ be an $n_1 \times n_2$ matrix and $\mathbf{D}$ be a $n_1 \times n_2 \times n_1 \times n_2$ tensor storing the vertex-to-vertex and edge-to-edge dissimilarities, respectively. The graph matching optimization problem is expressed in Eq 2, which takes the form of the quadratic assignment problem (QAP):

$$\underset{X \in \mathcal{X}}{\text{minimize}} \quad \sum_{i=1}^{n_1}\sum_{j=1}^{n_2}\sum_{k=1}^{n_1}\sum_{l=1}^{n_2} d_{ijkl} x_{ij} x_{kl} + \sum_{i=1}^{n_1}\sum_{j=1}^{n_2} c_{ij} x_{ij}. \tag{2}$$

Graphs are limited in their expressive power as edges can only relate pairs of vertices at a time; hypergraphs extend the definition of a graph to include hyperedges which can specify relationships among an arbitrary number of vertices. Hypergraph matching then concerns finding an optimal vertex correspondence between pairs of attributed hypergraphs. The number of vertices aligned by the most comprehensive hyperedge defines the degree of a hypergraph.

Maximum degree hypergraphs with hyperedges composed of all $n_1$ vertices yield the most comprehensive point-set matching function possible. The optimization objective function captures the dissimilarity arising between the matching: $(l_1, l_2, \ldots, l_{n_1}) \mapsto (l_1', l_2', l_3', \ldots, l_{n_1}')$. Then, for a given assignment matrix $X \in \mathcal{X}$, the hypergraph matching objective can be expressed using $n_1$ dissimilarity tensors of dimension 2, 4, . . ., 2d, . . ., $2n_1$, each measuring dissimilarity between degree $d$ hyperedges, respectively. Define $\mathbf{Z}^{(d)}$ as the tensor mapping the dissimilarity for the degree $d$ hyperedges. The hypergraph matching objective uses all $n_1$ hyperedge

dissimilarity terms, extending the degree 2 QAP objective:

$$f(X|\mathbf{Z}^{(1)}, \mathbf{Z}^{(2)}, \cdots, \mathbf{Z}^{(n_1)}) = \sum_{l_1=1}^{n_1} \sum_{l_1'=1}^{n_2} \mathbf{Z}_{l_1 l_1'}^{(1)} x_{l_1 l_1'} + \sum_{l_1=1}^{n_1} \sum_{l_1'=1}^{n_2} \sum_{l_2=l_1+1}^{n_1} \sum_{l_2'=1}^{n_2} \mathbf{Z}_{l_1 l_1' l_2 l_2'}^{(2)} x_{l_1 l_1'} x_{l_2 l_2'}$$

$$+ \sum_{l_1=1}^{n_1} \sum_{l_1'=1}^{n_2} \sum_{l_2=l_1+1}^{n_1} \sum_{l_2'=1}^{n_2} \sum_{l_3=l_2+1}^{n_1} \sum_{l_3'=1}^{n_2} \mathbf{Z}_{l_1 l_1' l_2 l_2' l_3 l_3'}^{(3)} x_{l_1 l_1'} x_{l_2 l_2'} x_{l_3 l_3'} + \ldots \qquad (3)$$

$$+ \sum_{l_1=1}^{n_1} \sum_{l_1'=1}^{n_2} \cdots \sum_{l_{n_1}=l_{n_1-1}+1}^{n_1} \sum_{l_{n_1}'=1}^{n_2} \mathbf{Z}_{l_1 l_1' \cdots l_{n_1} l_{n_1}'}^{(n_1)} x_{l_1 l_1'} \cdots x_{l_{n_1} l_{n_1}'}.$$

Hypergraph matching allows for the modeling of intricate point-set matching problems through high multiplicity assignment objective function formulations. The $\mathbf{Z}^{(d)}$ dissimilarity terms measure degree $d$ hyperedge dissimilarity comprising $d$ simultaneous vertex assignments. The range in assignment problem objective complexity from $d=1$ to $d=n_1$ trades off model capacity for increased computation. The traditional linear assignment problem ($d=1$) is solvable in polynomial time [4], but treats points between sets independently. Existing graphical methods ($d=2$) and hypergraphical methods ($d>2$) rely on approximate searches and do not generalize to high degree formulations of Eq 3. *Exact Hypergraph Matching* (*EHGM*) is able to find globally optimal solutions to hypergraph matching problems of arbitrary degree, allowing for the modeling of intricate point-set matching tasks.

## Related research

Finding an exact solution to the QAP is an $\mathcal{NP}$-hard problem. That is, unless *P=NP*, there does not exist a polynomial time solution to exactly solve the QAP [5]. Higher order assignment problems (i.e. hypergraph matching) are also $\mathcal{NP}$-hard as they are at least as hard as the QAP [6]. As a result, recent methods for graph matching and lower-degree hypergraph matching focus on heuristic solutions which offer no guarantee on performance [7–11]. Heuristic hypergraph matching methods are adapted from existing graph matching algorithms. In particular, spectral methods for solving graph matching (Eq 2) have been extended to solve hypergraph matching. Duchenne et al. [9] adapt Leordeanu's [1] work to obtain a rank-1 approximation of the affinity tensor via higher order power iteration. However, calculating affinity tensors ($\mathbf{Z}^{(d)}$ terms) is computationally prohibitive, especially for higher degree hypergraphs due to the exponentially growing number of entries in the tensors. Simplifying assumptions such as super-symmetry and sparseness are used with sampling methods to build large affinity tensors [9, 12]. Chertok and Keller propose similar methodology to [9], but instead unfold the affinity tensor and use the leading left singular vector to approximate the adjacency matrix [10]. All such methods operate outside the permutation matrix space. The Hungarian algorithm or similar binarization step is used to yield a valid assignment, e.g. as in [1].

Exactness allows for a more rigorous analysis of a hypergraphical point-set matching model than is possible using heuristic techniques. The guarantee of a globally optimal correspondence allows an iterative tuning of the underlying model in pursuit of accurate characterization, whereas the output of a heuristic algorithm could be incorrect due either to the stochasticity of the search or to inadequacy of the point-set matching model. Branch-and-bound is a paradigm originally developed to exactly solve the travelling salesman problem, a type of QAP [13, 14]. Branch-and-bound methods recursively commit partial assignments and solve successive subproblems within $\mathcal{X}$. The paradigm iteratively partitions the search space while bounding the optimum at each branch. At each step the method prunes branches which cannot contain the optimum. Convergence occurs when only feasible assignments achieving a

global optimum remain. The $\mathcal{NP}-$ hardness of the QAP implies convergence occurs only after implicit enumeration of $\mathcal{X}$.

## Overview of *EHGM* & application to *C. elegans*

*EHGM* deviates from recent graph matching and hypergraph matching methodology as an exact method, guaranteeing convergence to a globally optimal solution (S1 File: *Convergence of EHGM*). The algorithm builds upon the seminal branch-and-bound paradigm by extending the methodology to branch and prune based upon a given hypergraphical model [13]. A *k*-tuple of nodes at branch *m* is greedily selected while another step encapsulates the full hypergraphical objective upon selection. The decomposition of the objective into lower degree hyperedges used for steering the search and higher degree hyperedges fully evaluating branch decisions ensures completeness and enables flexibility in altering the underlying hypergraphical model.

Exactness limits *EHGM* to smaller problems (n $\leq$ 20); however, point-set matching tasks featuring larger numbers of points may be able to leverage lower degree relationships as points are ideally closer and relationships are less variant frame-to-frame. Sparsely sampled points may require an intricate model to adequately match; *EHGM* is applicable in the niche comprised of smaller point-sets requiring added context to adequately match. *EHGM* is better suited than competing heuristic methods for such smaller challenging point-set matching tasks for two reasons. First, the objective decomposition is independent of the branch-and-bound paradigm used to solve the optimization problem. As a result, point-set matching models of varying intricacies can be rigorously compared without substantial change to the algorithm. This feature allows for comparing models without having to use different methods. Second, *EHGM* is capable of solving maximum degree hypergraph matching problems, a first in the literature. The capability to jointly evaluate a complete alignment of points may enhance matching accuracy in challenging applications featuring smaller point-sets.

*EHGM* was inspired by *Caenorhabditis elegans* (*C. elegans*), a small, free-living roundworm often studied as a model of nervous system development due to its relative simplicity [15, 16]. The complete embryonic cell lineage has also been determined [17]; methods and technology have been developed to allow study of cell position and tissue development in the embryo [18–23]. However, the onset of muscular twitching in late-stage development invalidates traditional methods applied to track cells in the embryo. Recently developed methods use *seam cells* as fiducial markers to mitigate cellular displacement due to embryo repositioning [24].

The 20 seam cells and two associated neuroblasts form in lateral pairs along the left and right sides of the worm, resulting in eleven pairs upon hatching [17]. The neuroblasts appear in the final hours of development, just prior to hatching. The pairs of cells are named, anterior to posterior: *H0*, *H1*, *H2*, *V1*, *V2*, *V3*, *V4*, *V5*, *Q* (neuroblasts), *V6*, and *T*. Each pair's left and right cell is named accordingly; for example, *H1L* and *H1R* cells comprise the *H1* pair. We define *posture* as the identification of all seam cells and neuroblasts, which together reveal the shape of the coiled embryo. Fig 1A depicts locations of seam cell nuclei in an example image volume (left) and straightened to reveal the bilateral symmetry in seam cell locations (right). The process allows for studying tissue development despite late-stage twitching [24]. Capturing the image volumes at the desired spatial resolution necessitates pausing (5 minute interval) between image volumes to preserve embryo health. Repositioning between images contributes to the challenge in posture identification; Fig 1B shows four sequential images of an embryo, five minutes between images.

Current methods for posture identification rely on trained users to manually annotate the imaged nuclei using Medical Imaging, Processing, Analysis and Visualization (MIPAV), a 3D

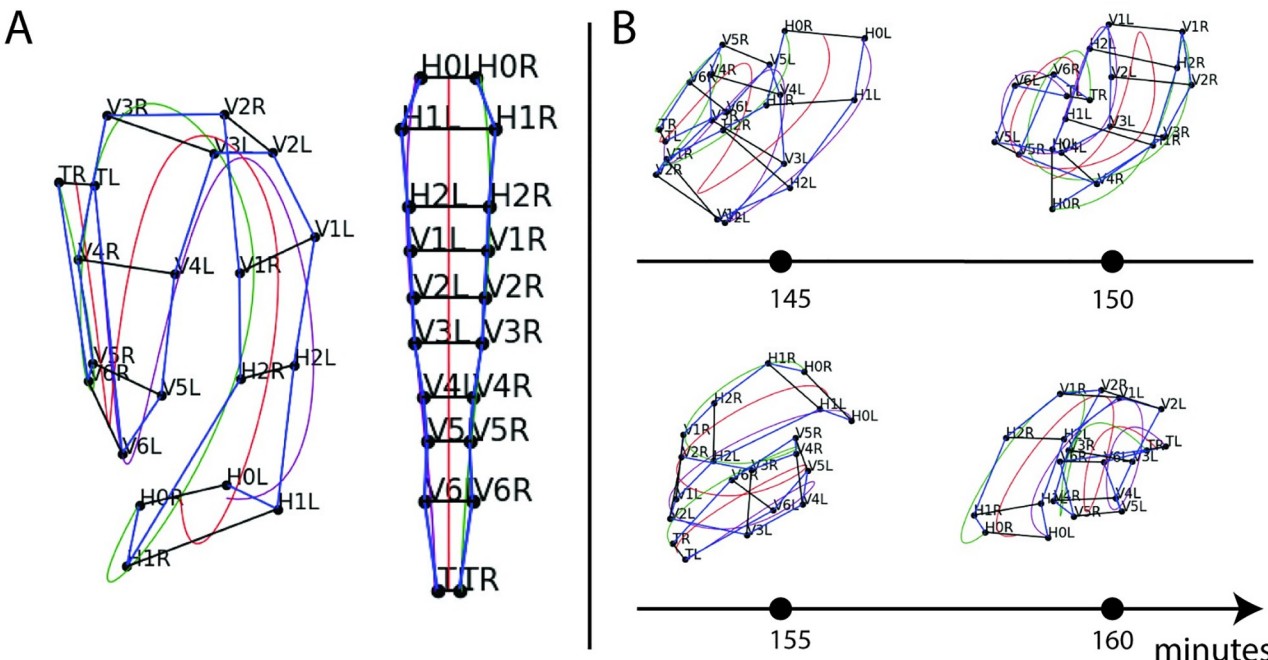

**Fig 1. Seam cells are used to recover *C. elegans* posture in high spatial resolution, low temporal resolution imaging.** A: Manually identified seam cell nuclei from an imaged *C. elegans* embryo. The cells form in pairs; they are labelled anterior to posterior: *H0*, *H1*, . . ., *V6*, *T*. The identification of all seam cells reveals the embryo's posture. Natural cubic splines through the left and right-side seam cells estimate the coiled embryo's body. The left image depicts identified nuclei connected to outline the embryo; splines are used to *untwist* the embryo, generating the remapped straightened points in the diagram on the right. B: Labelled nuclear coordinates from a sequence of four images. The embryo repositions in the five minute intervals between images, causing failure of traditional point-set matching approaches.

rendering tool [25]. The process takes several minutes per image volume and must be performed on approximately 100 image volumes per embryo [24]. Fig 2 depicts manually identified postures in the first two successive image volumes of Fig 1B. Manual annotation strategies leverage the bilateral symmetry and patterned shape of the embryo (thinner tail leading to a thicker body). Specks of fluorescence on the skin and other subtle visual cues not visible in Fig 2 often assist cell identification in challenging body positions. Posture identification serves as an intermediate step to late-stage cell tracking.

Posture identification allows for traditional frame-to-frame tracking of imaged cells belonging to various tissues such as the gut, nerve ring, and bands of muscle [24]. Images are captured in five minute intervals (Fig 1B) in order to achieve necessary resolution to track cells of other tissues without disturbing embryo development. For example, Fig 3A highlights muscle cell nuclei (red dots) with the identified seam cell nuclei (bold black dots) to contextualize the embryo's positioning. The posture is used to remap the muscle cells such that traditional cell tracking approaches can be applied in the late-stage embryo (Fig 3B). Fig 3C depicts the cell remapping process which uses splines fitted to the seam cell nuclei coordinates along the left and right sides [24]. The *untwisted* cell positions can then tracked frame-to-frame using traditional point-set matching methods (Fig 3D). Manual posture identification stands as a barrier to such analyses; *EHGM* was developed to approach the task.

Patterns throughout the embryo's body motivated our hypergraphical matching models, as established methods for point-set matching failed to adequately capture the relationships between seam cells throughout myriad twists and deformations of the developing embryo.

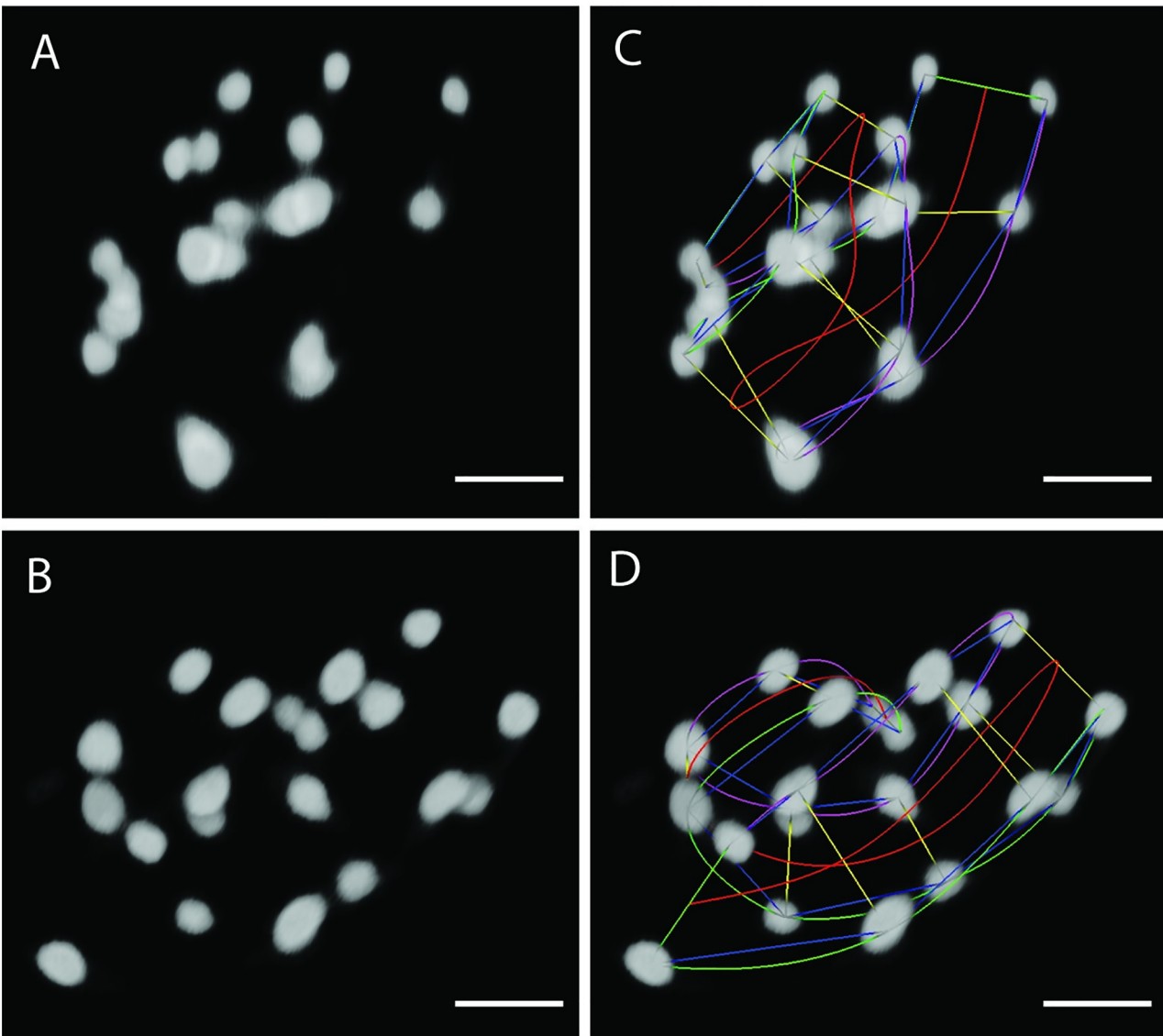

**Fig 2. Manual posture identification in two successive image volumes of Fig 1B using MIPAV.** The 20 fluorescently imaged seam cell nuclei rendered in two successive image volumes. Scale bar: 10 $\mu m$. A & B: Seam cell nuclei appearing in two successive image volumes visualized in MIPAV. The five minute interval allows the embryo to reposition between images, yielding entirely different postures. C & D: Manual seam cell identification by trained users reveals the posture. The curved lines are cubic splines as described in Fig 3C.

*EHGM* uses hypergraphical models comprising biologically driven geometric features to more accurately identify posture than established graphical methods. The limited expressive power of graphical models hinders accurate seam cell identification; graphical models accurately identify posture in 27% of samples compared to 56% using a hypergraphical model. User labelling of the posterior-most seam cell nuclei improves the success of hypergraph matching to correctly identifying all nuclei in 77% of samples. The improved accuracy in posture identification attributed to high-degree hypergraphical modeling solved via *EHGM* paves a path toward automatic posture identification while presenting a general framework for approaching similarly challenging point-set matching tasks.

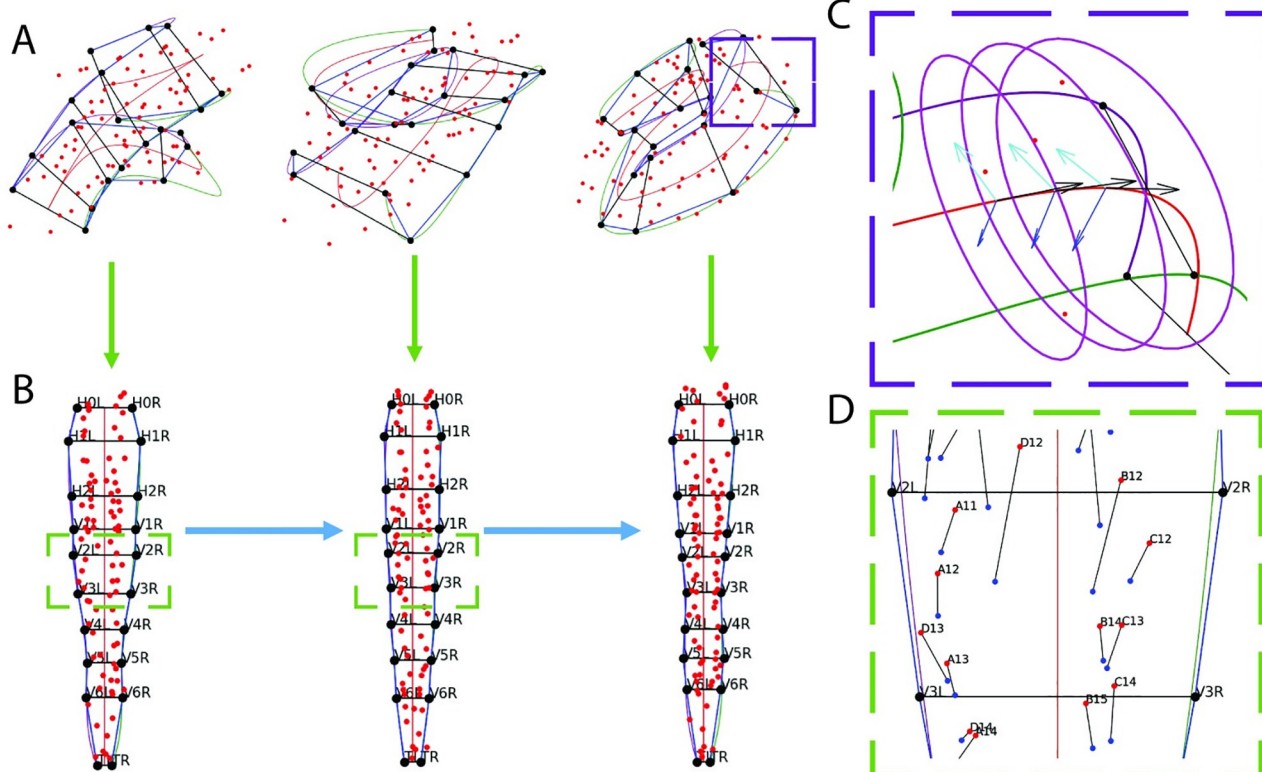

**Fig 3. Posture identification allows the tracking of other cells during late-stage embryogenesis.** A: Annotated seam cell nuclei coordinates (bold black) and muscle nuclei coordinates (red) from a sequence of three volumetric images. The untwisting process (green arrows) uses the seam cell locations to remap muscle cells to a common frame of reference. B: The remapped muscle nuclei are tracked frame-to-frame (blue arrows). C: A higher magnification view from the right coordinate plot of A. The left, right, and midpoint splines are used to create a change of basis defined by the tangent (black), normal (blue), and binormal (cyan) vectors. Ellipses are inscribed along the tangent of the midpoint spline, approximating the skin of the coiled embryo. D: A portion of the left (red) and center (blue) remapped muscle coordinates. Black lines connect the coordinates, frame-to-frame.

## Results

### Posture identification models

Posture was predicted via *EHGM* according to three models: a graphical model, denoted *Sides*, and two hypergraphical models: *Pairs* and *Posture*. The two hypergraphical models showcased *EHGM* as existing algorithms cannot find solutions under such high degree hypergraphs. Each of the three models used incrementally higher degree relationships to model posture. *Sides* followed the form of Eq 2, leveraging pairwise assignments to calculate lengths of portions of the embryo. *Pairs* used degrees four and six hyperedges to better model local regions of the embryo than is possible with graphical methods. *Posture* further demonstrated the capabilities of *EHGM* by including a degree $n_1$ hyperedge to maximize context in evaluating a hypothesized posture. Geometric features such as pair-to-pair rotation angles and left-right flexion angles were developed to measure and compare posture hypotheses more accurately. The calculation of each angle or distance requires identification of multiple seam cells in tandem to calculate, necessitating the use of edges and hyperedges.

Fig 4 depicts four types of models applied to perform posture identification on the two sampled images in Fig 2. Linear models (Fig 4A & 4B) are ill-equipped to identify posture due to the repositioning of the embryo between successive images; linear models were not applied to sample data. The graphical model *Sides* (Fig 4C & 4D) associates pairs of seam cells via edges

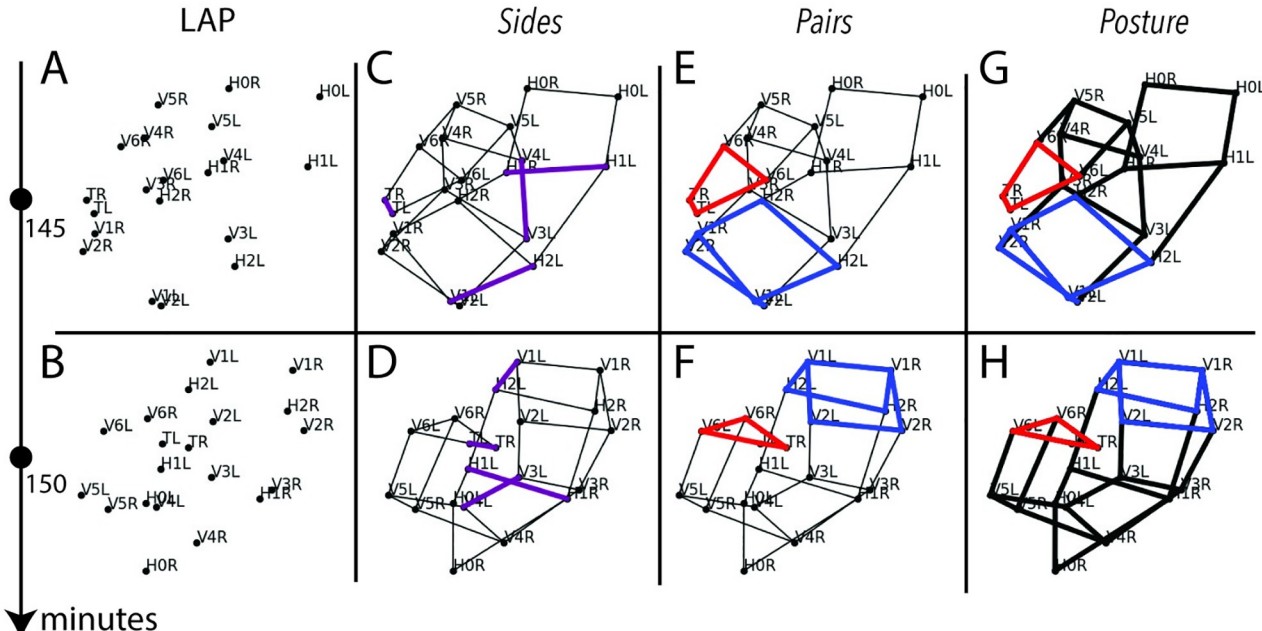

**Fig 4. Posture identification applied to the two successive images in Fig 2 according to a series of increasingly intricate models.** The embryo repositions between images. A & B: Linear models (LAP) cannot quantify relationships between seam cells; posture identification is impossible without context of neighboring cell identities. C & D: A graphical model (*Sides*) specifies edges (purple) between pairs of seam cell nuclei. Edge lengths are relatively static frame-to-frame, but the similarity of edge lengths throughout the embryo causes the edges to have a weak signal in identifying seam cells. E & F: The *Pairs* model uses degrees four (red) and six (blue) hyperedges to model a greater local context than is possible in a graphical model. G & H: The *Posture* model extends the *Pairs* model to use a degree $n_1$ (black) hyperedge to evaluate all seam cell assignments jointly.

(purple). Edge-wise features, lengths between adjacent cells, varied when the embryo coiled differently in successive images, but were otherwise approximately static frame-to-frame. However, the similarity in length measurements throughout the embryo yielded a model incapable of differentiating portions of the embryo (S1 Fig). Hypergraphical models *Pairs* (Fig 4E & 4F) and *Posture* (Fig 4G & 4H) used aforementioned hyperedges to more strongly characterize embryonic posture. Pair-to-pair hyperedges (red), three-pair sequences (blue), and $n_1$ degree hyperedges (black) allow for measuring angles and lengths more consistent frame-to-frame (S2 and S3 Figs).

## Posture identification accuracy

Annotators curated a dataset of seam cell nuclei center coordinates from 16 imaged embryos. Each imaged embryo yielded approximately 80 image volumes for a total of *N*=1264 labelled seam cell nuclei coordinate sets. Homogeneity in *C. elegans* embryo development allowed use of samples spanning multiple embryos to fit models via a leave-one-out approach (S1 File: *Model Fitting*, S1 File: *Posture Modeling*). EHGM allows for known correspondences, henceforth referred to as seeds, to be given as input prior to search initialization. The algorithm was evaluated both in a traditional point-set matching scenario given no *a priori* information, and in a series of seeded simulations. Seeded trials assumed incrementally more pairs given sequentially from the tail pair, *T*. KerGM [8], the leading algorithm for heuristic graph matching, was applied to posture identification using the same connectivity matrix as *Sides*. However, *KerGM* processed results frame-to-frame serially; this approach relied on using the correct posture at the prior image as input.

*EHGM* is able to store complete assignments encountered during the search as it compares against the current solution upon committing a final branch. This feature allowed for an analysis of the similarity between cost minimizing posture hypotheses and progressively higher cost solutions encountered during search. The necessity to identify *all* seam cells to form the posture motivated the implementation of a strict metric for success. The top *x* accuracy is defined as the percentage of all *N* samples in which *EHGM* returned the correct posture in the *x* lowest cost solutions; e.g. the top 1 accuracy describes the percentage of samples in which the correct posture was returned as the cost minimizing assignment, and the top 3 accuracy is the percentage of samples in which the correct posture was among 3 lowest cost posture hypotheses returned by the search.

Table 1 shows the percentage of all samples in which the correct posture (correct identification of *all* $n_1$ seam cells) was returned as the minimizer according to *KerGM* and each of the three models solved via *EHGM*: *Sides*, *Pairs*, and *Posture*. *KerGM* identified 27% of sampled postures correctly, outperforming *Sides* (10%). *Pairs* and *Posture* more effectively identified posture with 52% and 56% top 1 accuracies, respectively. *Pairs* and *Posture* achieved statistically similar average top 1 accuracies (independent two-sample pooled t-test, *p*=0.051, one-sided).

Differences between reported top accuracies, particularly in *Pairs* and *Posture* model results, reflect the challenge in posture identification. Hypergraphical models' objective values for correct postures were often similar for hypotheses with minor mistakes. Notably, the *Posture* model returned the correct posture in the top 3 hypotheses in approximately 67% of samples, an approximate 20% increase in relative accuracy over the top 1 percentage, 56%.

Posture identification results were stratified by the presence of the *Q* neuroblasts; 875 of the 1264 samples contain only the seam cells while the remaining 389 samples were developed enough to express the *Q* neuroblasts as well. Table 2 depicts the findings presented in Table 1 split by *Q* neuroblast presence. All methods achieved a higher accuracy on post-*Q* samples. The stratification revealed an advantage of the *Posture* model over *Pairs* for the pre-*Q* samples: 48% vs 44% (independent two-sample pooled t-test, *p*=.034, one-sided). The hypergraphical models' enhanced top 1 accuracies on the post-*Q* samples, 71% vs. 44% and 72% vs. 48%,

**Table 1. Hypergraphical models *Pairs* and *Posture* achieved highest posture identification accuracy.** Posture identification accuracies across all *N*=1264 samples. Point-set matching models are listed across columns: *KerGM* [8] was compared to proposed *EHGM* models. Rows list the top *x* accuracy as a percentage of samples. The differences between top *x* accuracies across hypergraphical models highlight the difficulty in posture identification.

| Accuracy | KerGM [8] | Sides | Pairs | Posture |
|----------|-----------|-------|-------|---------|
| Top 1 | 0.27 | 0.10 | 0.52 | 0.56 |
| Top 2 | 0.27 | 0.14 | 0.60 | 0.65 |
| Top 3 | 0.27 | 0.15 | 0.63 | 0.67 |

**Table 2. Hypergraphical models leveraged *Q* neuroblasts to more accurately identify posture.** The samples were split according to the absence (left) or presence (right) of the *Q* neuroblasts, which form in the last two hours of development. There were 875 $n_1$=20 cell (pre-*Q*) samples and 389 $n_1$=22 cell (post-*Q*) samples. Hypergraphical models *Pairs* and *Posture* more accurately identified posture in the post-*Q* samples than the pre-*Q* samples, suggesting the increased continuity along the body enhanced posture modeling.

| Accuracy | Pre-Q ($n_1 = 20$) | | | | Post-Q ($n_1 = 22$) | | | |
|----------|-----------|-------|-------|---------|-----------|-------|-------|---------|
| | KerGM [8] | Sides | Pairs | Posture | KerGM [8] | Sides | Pairs | Posture |
| Top 1 | 0.25 | 0.07 | 0.44 | 0.48 | 0.35 | 0.19 | 0.71 | 0.72 |
| Top 2 | 0.25 | 0.10 | 0.51 | 0.57 | 0.35 | 0.25 | 0.80 | 0.81 |
| Top 3 | 0.25 | 0.11 | 0.55 | 0.60 | 0.35 | 0.26 | 0.82 | 0.82 |

**Table 3. Hypergraphical models more aptly contextualized posture than a graphical model but required more computation.** Runtime (minutes) refers to the median runtime of each model in minutes. Cost ratio reports the median cost ratio, defined as the ratio of the correct posture cost to the returned posture cost. Hypergraphical models more effectively described posture than the graphical model at expense of computation.

| | Pre-Q ($n_1 = 20$) | | | Post-Q ($n_1 = 22$) | | |
|---|---|---|---|---|---|---|
| | *Sides* | *Pairs* | *Posture* | *Sides* | *Pairs* | *Posture* |
| Runtime (minutes) | 4.81 | 34.25 | 51.12 | 9.66 | 56.58 | 72.60 |
| Cost Ratio | 1.36 | 1.04 | 1.00 | 1.16 | 1.00 | 1.00 |

demonstrates the advantage of hypergraphical modeling for posture identification. The increased continuity of the embryo via the Q neuroblasts provided substantial context in defining the coiled embryo by better penalizing incorrect postures.

We then investigated how effectively each of the proposed models captured patterns in posture identification using a cost ratio metric. The cost ratio was defined as the ratio of the correct posture's objective value to the cost minimizing posture's objective value for each model. A cost ratio greater than one implied the hypothesized posture's objective value of was lower than that of the correct posture, suggesting the model did not aptly characterize posture as an incorrect posture hypothesis was preferred by the model. Table 3 highlights the median runtimes (minutes) and median cost ratios across samples split by Q cell presence. *Posture* outperformed *Pairs* on the pre-Q samples not just according to top 1 accuracy, but also cost ratio (1.00 vs. 1.04, two-sample Kolmogorov-Smirnov (KS) test, $p=0.027$, one-sided). The distinction between models did not extend to the post-Q samples, echoing their similar top 1 accuracies (1.00 vs. 1.00, two-sample KS test, $p=0.20$, one-sided). Nevertheless, hypergraphical modeling (represented by *Posture*) modeled posture more effectively than graphical modeling (represented by *Sides*) on both pre-Q (1.00 vs. 1.36, two-sample KS test, $p<10^{-72}$, one-sided) and post-Q (1.00 vs. 1.16, two-sample KS test, $p<10^{-51}$, one-sided) samples. Improvements on posture identification performance attributed to hypergraphical modeling came at the cost of increased computation. The the addition of the $n_1$ degree hyperedge in *Posture* increased runtime over the *Pairs* model on pre-Q samples (51.12 minutes vs. 34.25 minutes, two-sample KS test, $p<10^{-22}$, one-sided).

Seeded experiments specifying nuclear identities provided *a priori* information starting with the tail pair (*T*), and incrementally identified more pairs (*V6*, *V5*, etc.). Each experiment was given four hours of maximum runtime. Table 4 reports top accuracies for *EHGM* models by number of seeded pairs. Seeding the especially challenging posterior region greatly improved top accuracies, but not enough to fully solve posture identification. Particularly, *Posture* showed a 15% improvement in top 1 accuracy (54% to 62%) when seeded with the tail pair, *T*. On the other hand, *Sides* did not impactfully improve performance with tail pair seeding. Further seeding improved results for all models. Fig 5 depicts top 1 accuracies (red) and median runtimes (blue, log-scaled) across seeded experiments for the *Sides* (dashed

**Table 4. Seeding posterior pair identities enabled more accurate posture identification.** Top accuracies are reported across all samples on each row; each simulation had a four hour maximum runtime. Spanning columns specify which model was used while subcolumns under each model list which pairs were given as seeds prior to search. The *None* columns recreate the original no information case reported in Table 1, but with the runtime limit.

| Accuracy | Sides | | | | Pairs | | | | Posture | | | |
|---|---|---|---|---|---|---|---|---|---|---|---|---|
| | None | T | T-V6 | T-V5 | None | T | T-V6 | T-V5 | None | T | T-V6 | T-V5 |
| Top 1 | 0.10 | 0.11 | 0.22 | 0.28 | 0.53 | 0.59 | 0.72 | 0.78 | 0.54 | 0.62 | 0.72 | 0.78 |
| Top 2 | 0.14 | 0.15 | 0.27 | 0.34 | 0.60 | 0.64 | 0.76 | 0.81 | 0.63 | 0.67 | 0.77 | 0.82 |
| Top 3 | 0.15 | 0.16 | 0.27 | 0.35 | 0.63 | 0.66 | 0.77 | 0.82 | 0.64 | 0.68 | 0.78 | 0.82 |

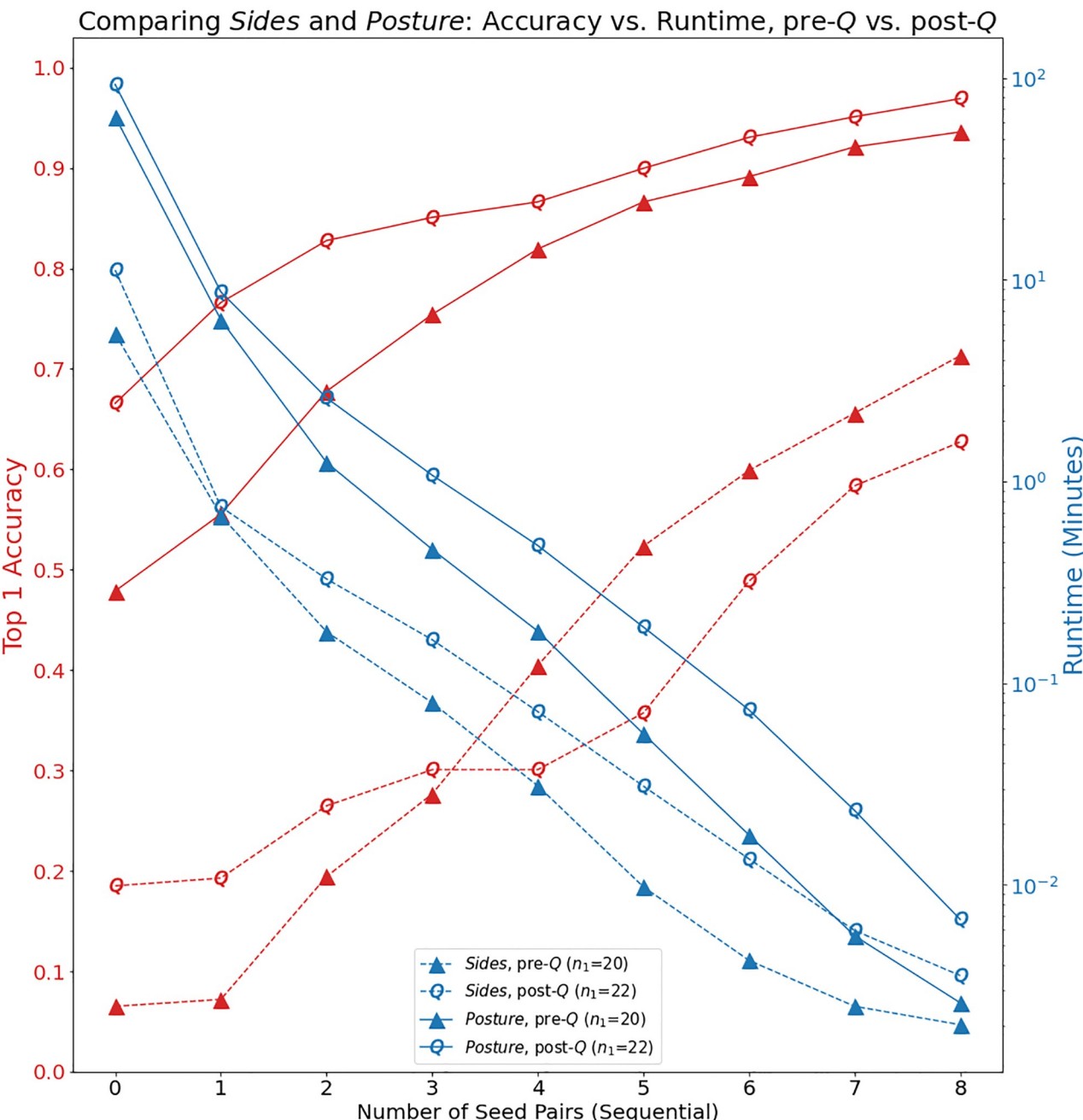

**Fig 5. Posterior region seeding improved posture identification in both graphical (*Sides*) and hypergraphical (*Posture*) models while exponentially decreasing runtime.** The *Sides* (dashed lines) and *Posture* (solid lines) models are compared by *Q* pair presence (pre-*Q*: triangles, post-*Q*: *Q*s). Top 1 accuracies (red) improved across models while runtimes (blue) especially fell with *T* pair seeding (1 on the horizontal axis). Hypergraphical modeling especially benefited from posterior pair seeding while *Sides* required more context to improve.

lines) and *Posture* (solid lines) models split by *Q* pair presence (pre-*Q*: triangles, post-*Q*: *Q*s). *Posture* leveraged *T* pair seeding to reduce median runtime from 51 minutes to 6 minutes and 72 minutes to 9 minutes while improving top 1 accuracy to 56% and 76%, pre-*Q* and post-*Q*, respectively.

## Discussion

We have presented *EHGM* as a dynamic and effective tool for intricate point-set matching tasks. The hypergraph matching algorithm provides a method in which to gauge the efficacy of modeling point correspondences in conservatively sized problems; problems featuring larger numbers of points likely contain the context required to adequately match via lower degree models. For example, post-*Q* samples were more accurately identified across models, but the largest marginal gain in accuracy came from *Sides* (19%) to *Pairs* (71%). The results suggest that added context throughout the embryo would further improve posture identification accuracy, reducing the reliance on higher degree (and thus more computationally expensive) hypergraphical models. *EHGM* specifically addresses a gap in literature concerning challenging point-set matching applications in which domain-specific features lead to rigorously testable models. Seeding allows a wider range of problems to be approached by mitigating computational expense of the algorithm for scenarios featuring larger point-sets.

Posture identification in embryonic *C. elegans* is a challenging problem benefiting from high degree hypergraphical modeling. *EHGM* equipped with biologically inspired hypergraphical models led to substantial improvement in posture identification. The top 1 accuracy doubled from 27% with a graphical model to 56% via the *Posture* model (Table 1). The top 3 accuracy rate improved to 67%, highlighting the challenge in precisely specifying the coiled embryo due to the similarity of competing posture hypotheses. The presence of *Q* neuroblasts further contributed to accurate posture identification. The added context empowered the *Posture* model to identify the correct posture in 82% of post-*Q* samples (Table 2).

The top *x* percentage accuracy metric reflects the need to correctly identify *all* seam cells to recover the underlying posture but does not distinguish between hypotheses that are incorrect due to one cell identity swap or a more systemic modeling inadequacy. A qualitative analysis highlighted a few themes among incorrectly predicted postures. The foremost errors concern the tail pair cells, *TL* and *TR*; spurious identifications occurred when the tail pair coiled against another the body of the embryo, causing one tail cell identity to be interchanged with a cell of a nearby body pair. The variance of feature measurements in the posterior region resulted in similar costs for postures with minor differences about the posterior region.

Pair seeding allowed for the strengths of *EHGM* to compensate for the most challenging aspect of posture identification. The posterior region of the embryonic worm is especially flexible and contributed to most of the incorrectly predicted postures. Feature engineering stands to create hypergraphical models more capable of reliable posture identification, particularly in contextualizing the posterior region. The method and application outline a protocol for challenging point-set matching tasks.

## Methods

### Exact Hypergraph Matching

*EHGM* extends the branch-and-bound paradigm to exactly solve hypergraph matching. The algorithm performs the search in the permutation space $\mathcal{X}$ subject to a given branch size $k$ which specifies the number of vertices assigned at each branch. A size $n_1$ hypergraph will require $M := \frac{n_1}{k}$ branch steps, where branch $m$ concerns the assignment of vertices $((m-1)k + 1, (m-1)k + 2, \ldots, mk)$; vertices $1, 2, \ldots, mk$ have been assigned upon completion of the $m^{th}$ branch. The set $\mathbf{P}$ contains all possible permutations of the indices of the unordered point set, $|\mathbf{P}| = \frac{n_2!}{(n_2-k)!}$. $\mathbf{P}$ is incrementally subset into queues $\mathbf{Q}_m \subseteq \mathbf{P}$ at branches $m = 1, 2, \ldots, M$ at each branching. The queue $\mathbf{Q}_m$ is subset according to both a pruning rule which eliminates

permutations leading to a suboptimal solution as well as the one-to-one constraints of $\mathcal{X}$. The search converges to a global optimum upon the implicit enumeration of $\mathbf{Q}_1 = \mathbf{P}$.

The objective function $f$ is further stratified based upon the branch size $k$. Lower degree ($d \leq 2k$) hyperedge dissimilarity tensors are computed prior to search. Branches comprising $k$-tuples of vertices are partially assigned in a greedy manner using the lower degree hyperedge dissimilarities via the selection rule $H$. Later branches accrue higher degree ($d > 2k$) hyperedge dissimilarities which are calculated at time of branching; the intent of the method is to rely on lower degree terms to steer the search towards an optimum in effort to minimize the number of branches explored. The aggregation rule $I$ accrues higher degree hyperedge dissimilarity terms upon branching, further guiding the pruning step and ensuring the complete specification of the objective $f$.

The branching and selection rules are designed to reduce computation performed throughout the search. A partial assignment at branch $m$: $\mathbf{K}_m = (l'_{(m-1)k+1}, l'_{(m-1)k+2}, \ldots, l'_{mk}) \in \mathbf{Q}_m$ is selected via precomputed lower degree hyperedge dissimilarity tensors $\mathbf{Z}^{(1)}, \ldots, \mathbf{Z}^{(2k)}$. A larger branch size $k$ results in a selection rule with larger scope of the optimization landscape, better equipped to place optimal branches earlier in each queue $\mathbf{Q}_m$ at time of branching. However, computing the lower degree dissimilarity tensors prior to search can be prohibitively expensive for larger point-sets.

## Selection & aggregation

The first branch permutation $\mathbf{K}_1 = (l'_1, l'_2, \ldots, l'_k) \in \mathbf{Q}_1 = \mathbf{P}$ assigns vertices $(l_1, l_2, \ldots, l_k)$ to points $(l'_1, l'_2, \ldots, l'_k)$ according to the initial branch selection rule $H_1$ (Eq 4). $H_1$ defines a cost given dissimilarity tensors $\mathbf{Z}^{(1)}, \mathbf{Z}^{(2)}, \ldots \mathbf{Z}^{(k)}$ according to a permutation $\mathbf{K}_1$. The $k$ pairs of constraints given by the branch $m$ and permutation of point indices $\mathbf{K}_m$: $\{(l_1, l'_1), \ldots, (l_k, l'_k)\}$ enables a simplification in the objective formulation. $H_1$ can be simply described as quantifying the first $k$ assignment costs for hyperedge degrees 1 to $k$:

$$H_1(\mathbf{K}_1 | \mathbf{Z}^{(1)}, \mathbf{Z}^{(2)}, \ldots, \mathbf{Z}^{(k)}) :=$$
$$\sum_{i_1=1}^{k} \mathbf{Z}^{(1)}_{l_{i_1} l'_{i_1}} + \sum_{i_1=1}^{k} \sum_{i_2=i_1+1}^{k} \mathbf{Z}^{(2)}_{l_{i_1} l'_{i_1} l_{i_2} l'_{i_2}} + \ldots + \sum_{i_1=1}^{k} \sum_{i_2=i_1+1}^{k} \cdots \sum_{i_k=i_{k-1}+1}^{k} \mathbf{Z}^{(k)}_{l_{i_1} l'_{i_1} l_{i_2} l'_{i_2} \cdots l_{i_k} l'_{i_k}}. \tag{4}$$

Subsequent branches $m = 2, 3, \ldots M$ then use the general selection rule $H_m$ to order the permutations of the $m^{th}$ branch: $\mathbf{K}_m = (l'_{(m-1)k+1}, l'_{(m-1)k+2}, \ldots l'_{mk}) \in \mathbf{Q}_m$. Branch $m$ incurs a selection rule cost $H_m$ (Eq 5) comprising lower degree hyperedge dissimilarities for assignments both within branch $m$ and the assignments between branches $1, 2, \ldots, m-1$ and branch $m$. The partial assignment constraints $\mathbf{K}_m$ allow further simplification of notation; the reversed order of summation indices satisfies the criteria that only hyperedge dissimilarities pertaining to branch $m$ assignments are considered via $H_m$:

$$H_m(\mathbf{K}_m | \mathbf{K}_1, \ldots, \mathbf{K}_{m-1}, \mathbf{Z}^{(1)}, \ldots, \mathbf{Z}^{(2k)}) :=$$
$$\sum_{i_1=(m-1)k+1}^{mk} \mathbf{Z}^{(1)}_{l_{i_1} l'_{i_1}} + \sum_{i_2=(m-1)k+1}^{mk} \sum_{i_1=1}^{i_2-1} \mathbf{Z}^{(2)}_{l_{i_1} l'_{i_1} l_{i_2} l'_{i_2}}$$
$$+ \sum_{i_3=(m-1)k+1}^{mk} \sum_{i_2=1}^{i_3-1} \sum_{i_1=1}^{i_2-1} \mathbf{Z}^{(3)}_{l_{i_1} l'_{i_1} l_{i_2} l'_{i_2} l_{i_3} l'_{i_3}} + \ldots + \sum_{i_{2k}=(m-1)k+1}^{mk} \sum_{i_{2k-1}=1}^{i_{2k}-1} \cdots \sum_{i_1=1}^{i_2-1} \mathbf{Z}^{(2k)}_{l_{i_1} l'_{i_1} \cdots l_{i_{2k}} l'_{i_{2k}}}. \tag{5}$$

The greedy selection rule orders queues $\mathbf{Q}_m$ but does not account for higher degree ($2k < d \leq n_1$) hyperedge dissimilarities. Precomputing higher degree dissimilarity tensors can be both

computationally expensive and inefficient as ideally only a small percentage of combinations are queried throughout the search. The aggregation rule $I_m$ (Eq 6) measures the dissimilarity attributable to higher degree ($2k < d \leq mk$) hyperedges accessible due to branch $m$ partial assignments. The aggregation rule updates the cost of branch $\mathbf{K}_m$ assignments, further informing the pruning step to subset the next queue $\mathbf{Q}_{m+1}$. The greedy selection rule $H_m$ in tandem with the aggregation rule $I_m$ aim to minimize the total computation performed in finding an optimum. The definition $I_m$ follows the form of the general selection rule $H_m$ but applied to the higher degree hyperedge dissimilarities. The aggregation rule $I_m$ (Eq 6) can be expressed as the degree $d$ dissimilarities calculable upon assignments of branch $m$ assignments for degrees $2k < d \leq mk$:

$$I_m(\mathbf{K}_m | \mathbf{K}_1, \mathbf{K}_2, \ldots, \mathbf{K}_{m-1}, \mathbf{Z}^{(2k+1)}, \ldots, \mathbf{Z}^{(mk)}) :=$$

$$\sum_{d=2k+1}^{mk} \sum_{i_d=(m-1)k+1}^{mk} \sum_{i_{d-1}=1}^{i_d-1} \cdots \sum_{i_1=1}^{i_2-1} \mathbf{Z}_{l_{i_1} l'_{i_1} \ldots l_{i_d} l'_{i_d}}^{(d)}. \tag{6}$$

The $m$th branch allows for hyperedge dissimilarities up to degree $mk$ concerning the first $mk$ assignments. The $M$th branch yields a complete assignment, allowing the evaluation of maximum degree $n_1$ hyperedge dissimilarities. The partitioning and further regrouping of each $H_m$ and $I_m$ as defined fully accounts for the objective $f$ while allowing efficient computation during the search (S1 File: *Hypergraphical Objective Decomposition*, S1 File: *Convergence of EHGM*).

## Posture identification in embryonic *C. elegans*

*Caenorhabditis elegans* (*C. elegans*) is a small, free-living nematode found across the world. The worm is often studied as a model of nervous system development due to its relative simplicity [15–17, 26]. The adult worm features only 302 neurons, the morphology and synaptic patterning of which have been determined via electron microscopy [15]. The complete embryonic cell lineage has also been determined [17]; methods and technology have been developed to allow study of cell position and tissue development in the embryo [18–23]. Systems-level studies of these processes may be able to discover larger-scale principles underlying developmental events.

The embryo features a set of twenty *seam cells* and two associated neuroblasts. The seam cells and neuroblasts together describe anatomical structure in the coiled embryo, acting as a type of "skeleton" outlining its body. Identification of the seam cells and neuroblasts defines the embryo's posture. Fluorescent proteins are used to label cell nuclei, including the seam cell nuclei so that they may be visualized during imaging, e.g., with light sheet microscopy [27]. Volumetric images are captured at five minute intervals in order to capture subcellular resolution without damaging the worm's development [24]. Seam cell nuclei appear in the fluorescent images as homogeneous spheroids. Their positions relative to other nuclei and other salient cues present in the image volumes comprise the information that trained users employ to manually identify seam cells. Fig 6 shows the two rendered fluorescent images from Fig 1A in Medical Image Processing, Analysis and Visualization (MIPAV), a 3D rendering tool [25]. The interface is used to annotate both seam cell nuclei and track remapped nuclei, as in Fig 2 [24].

We cast posture identification as hypergraph matching and use *EHGM* to solve the resulting optimization problem. The proposed models: *Sides*, *Pairs*, and *Posture* trade off modeling capacity for increased computation to identify optimal solutions. *Sides* expresses posture identification as graph matching; edge-wise (degree $d$=2) features take the form of standardized

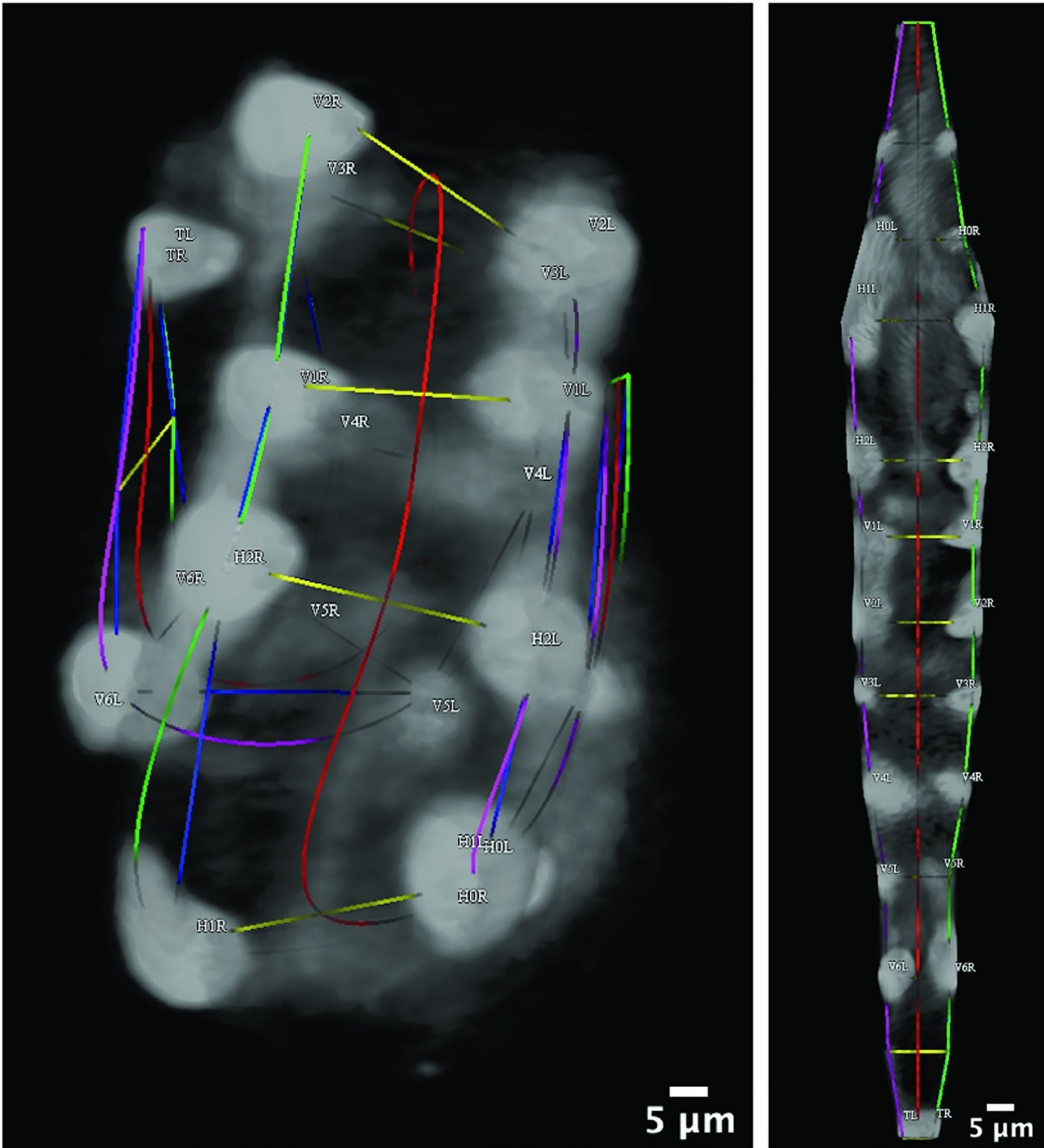

**Fig 6. Rendered image volumes in the MIPAV GUI.** The imaged twisted embryo (left) and imaged straightened embryo (right) rendered in Medical Image Processing, Analysis and Visualization (MIPAV) [25]. The fluorescent images are those depicted in Fig 1A. Trained users navigate the MIPAV GUI to identify seam cells based upon relative positioning and other salient features such as specks of fluorescence on the skin. Correct identification of all imaged nuclei reveals the coiled embryonic posture. Green (left), red (center), and purple (right) splines yield an approximation of the coiled embryo's posture. Yellow lines connect seam cell nuclei laterally. The splines are used with the image volume to sweep planes orthogonal to the center spline, yielding the straightened embryo image.

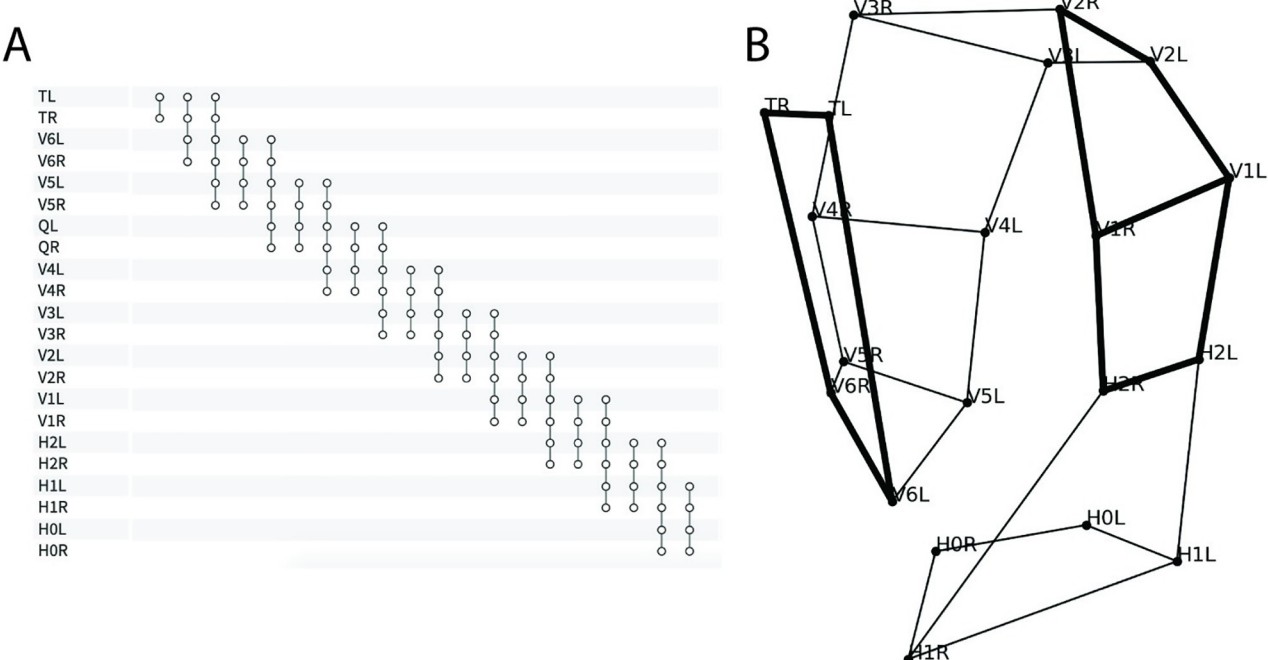

**Fig 7. The *Pairs* hypergraphical model uses expansive local contexts about each portion of the embryo.** A: The *Pairs* hyperedges connect local seam cell nuclei in sets of four and six. B: Degree four hyperedges connect sequential pairs of seam cells while degree six hyperedges connect sequential triplets of pairs. The posterior-most degree four hyperedge and a central degree six hyperedge are **bolded**.

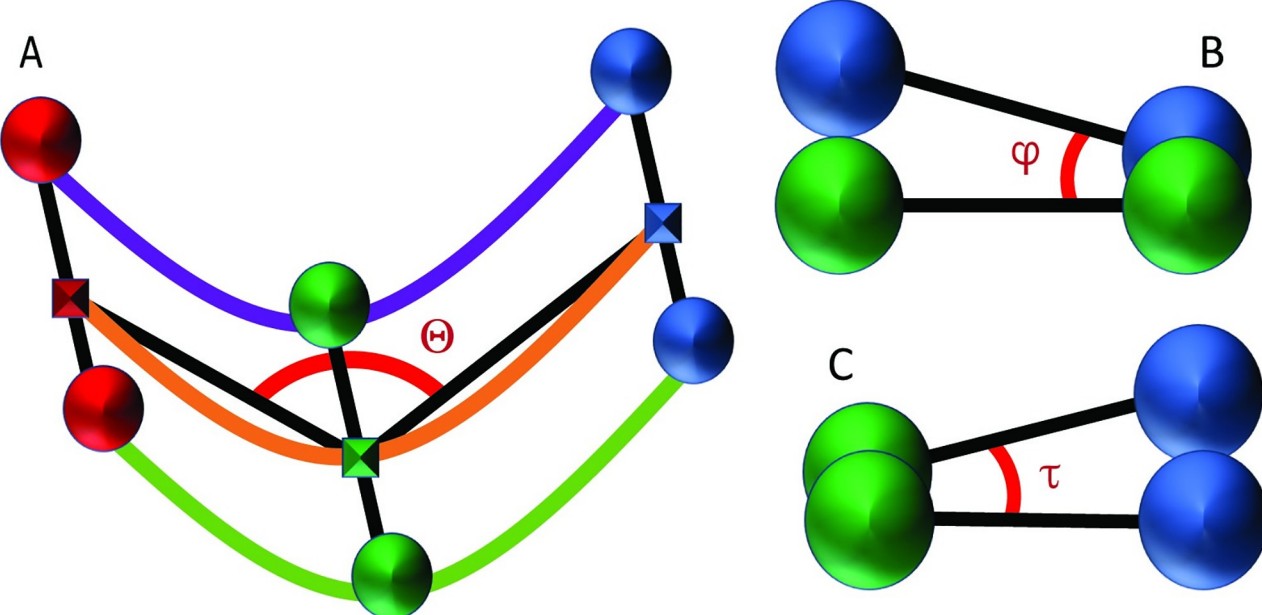

**Fig 8. Hypergraphical geometric features contextualize seam cell assignments.** Anatomically inspired geometric features describe bend and twist of a posture assignment. A: Three pairs of sequential nuclei: red, green, blue. Rectangles represent pair midpoints. The angle Θ in red is used as a degree six feature given six nuclei assignments. B, C: Degree four hypergraphical features measuring twist angles $\varphi$ and $\tau$. These angles measure posterior to anterior twist pair-to-pair and left-right twist, respectively.

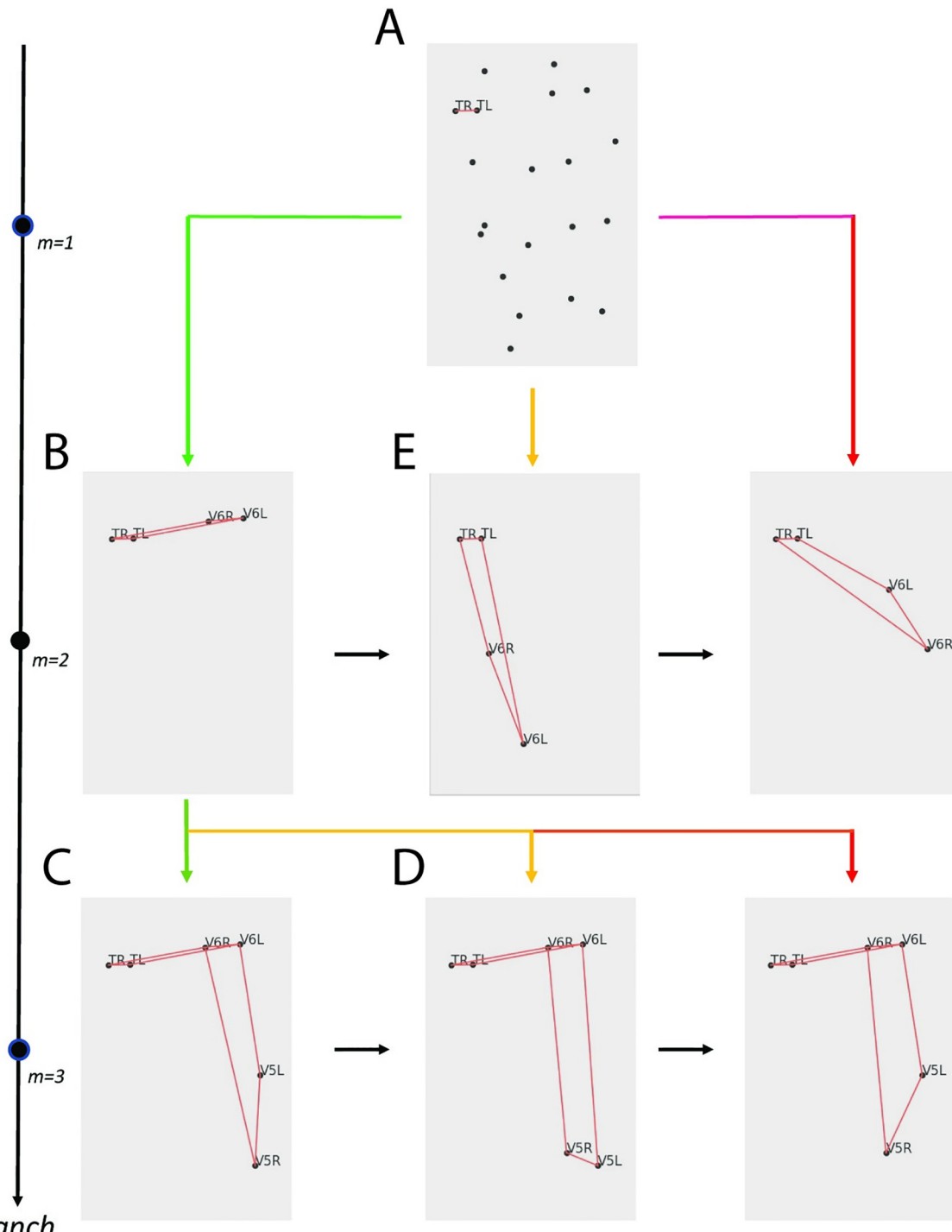

**Fig 9. *EHGM* applied to the sample image depicted in Fig 1A. A:** Two points are selected at the initial branch for *TL* and *TR*, respectively. Candidates for the successive pair, *V6L* and *V6R*, are queued based on hypergraphical relationships between the established cell identities *TL* and *TR* and each hypothesized *V6* pair (lower costs are green to higher costs in red). **B:** The leading hypothesis at branch *m*=2 given the initial branch pair is chosen. The recursion continues to queue *V5* pair choices at branch *m*=3. Black arrows within branch *m* specify the ordering of the branch given established cell assignments. Each branch creates a new subproblem of completing the posture given partially assigned identities. **C:** The tree continuing from the *V5* pair hypothesis is fully explored according to the established recursion. **D:** The next leading *V5* hypothesis is initiated upon exhaustion of the subtree formed at panel C. **E:** Implicit enumeration of the subtree formed at panel B causes the search to progress to the second leading *V6* hypothesis.

chord lengths between nuclei laterally and sequentially along each side. The first hypergraphical model, *Pairs*, employs a greater local context than *Sides* using degrees four and six hyperedges to describe relationships between seam cells. Hyperedges formed by two or three sequential pairs ($d$=4,6) better detail local regions throughout the embryo than is capable of a graphical model. Fig 7A presents the hyperedge connectivity among nodes in the *Pairs* model [28]. The *Posture* model extends the *Pairs* model by leveraging complete posture ($d=n_1$) features in effort to further discriminate between posture hypotheses that appear similar in sequential regions of the embryo. Geometric features help contextualize the coiled posture. Fig 8 illustrates three of the features used in the *Pairs* and *Posture* models. The angle Θ measures the angle between three successive pair midpoints. The angles Θ decrease throughout development as the worm elongates. Pair-to-pair twist angles $\varphi$ and $\tau$ penalize posture hypotheses in which posterior to anterior transitions are jagged and unnatural in appearance. See S1 File: *Posture Modeling* for further details and specification of model features.

The traditional point-set matching task requires a labelled point-set and a second unidentified point-set. Higher order features such as bend and twist angles may vary largely frame-to-frame depending on the posture at moment of imaging. However, elongation throughout late-stage development causes macroscopic trends in these geometric features. We estimate a template posture as a composite of feature measurements from a corpus of manually annotated postures. The templates are time dependent to reflect the elongation from the first point of imaging throughout development until hatching. See S1 File: *Model Fitting* for details on template estimation.

Together, the fitted models are used with *EHGM* to identify posture in imaged *C. elegans* embryos. The branch size $k$=2 is set for all models, i.e. a lateral pair of seam cell identities are assigned at each branch starting with the tail pair cells *TL* and *TR*. The successive pair cells, *V6L* and *V6R*, are assigned given the established cells and hypergraphical relationships accessible with the hypothesized identities. Fig 9 depicts *EHGM* applied to the sample image depicted in Fig 1A. The initial pair (*TL* and *TR*) is selected, instantiating a search tree (Fig 9A). Successive seam cell identities are partially assigned according to the given hypergraphical model in a pair-wise fashion. Each branch greedily queues hypothesized point-pair assignments conditioned on the previous branch assignments (black arrows within a branch). The next leading *V6* pair (Fig 9E) is chosen upon exhaustion of the leading hypothesized *V6* pair (Fig 9B). *EHGM* continues the recursion to implicitly identify a globally optimal posture under the given hypergraphical model; each possible initial pair will follow this illustrated process subject to pruning of the minimizing posture accessed via the hypothesized tail pair in Fig 9A.

## Supporting information

**S1 File.**
(PDF)

**S1 Fig. *Sides* model features.** A) Distances between nuclei of lateral pairs. Notably, the tail pair distance (left-most panel) is constant throughout imaging. The tail pair distance informs the initial pair selection rule $H_1$. B) Chord lengths along left and right sides of the posture. Both quadratic features show high variance.
(TIF)

**S2 Fig. *Pairs* model features.** A) Ratios of pair distances. B) Distance between successive pair midpoints. C) Cosine similarities between successive left and right sides. D) Lateral axial twist angles. E) Axial twist angles. F) Midpoint bend angles. G) Planar intersection angles.
(TIF)

**S3 Fig. *Posture* model features include all *Pairs* features and posture-wide versions of *Pairs* features.** A) Summed ratios of pair distances. B) Summed distances between successive pair midpoints. C) Summed cosine similarities between successive left and right sides. D) Summed lateral axial twist angles. E) Summed axial twist angles. F) Summed midpoint bend angles. G) Summed planar intersection angles.
(TIF)

# Acknowledgments

This work was supported by the Intramural research program of the National Institute of Biomedical Imaging and Bioengineering within the National Institutes of Health, and the Howard Hughes Medical Institute (HHMI). This article is subject to HHMI's Open Access to Publications policy. HHMI lab heads have previously granted a nonexclusive CC BY 4.0 license to the public and a sublicensable license to HHMI in their research articles. Pursuant to those licenses, the author-accepted manuscript of this article can be made freely available under a CC BY 4.0 license immediately upon publication. This work utilized the computational resources of the NIH HPC Biowulf cluster. (http://hpc.nih.gov). Dr. Evan Ardiel was instrumental in developing descriptive features for identifying worm posture. Post-Baccalaureate research fellows Brandon Harvey and Nensi Karaj were supportive in providing data and discussions concerning the modeling. Dr. Zhen Zhang and Dr. Arye Nehorai provided assistance in using *KerGM* [8]. Dr. Vincent Lyzinski also provided insight on the methods. We also thank Dr. Hank Eden and Dr. Matthew Guay for their careful readings and suggestions. The code and data are available at https://github.com/lauziere/EHGM.

# Author Contributions

**Conceptualization:** Andrew Lauziere, Hari Shroff.

**Data curation:** Ryan Christensen.

**Formal analysis:** Andrew Lauziere.

**Investigation:** Andrew Lauziere.

**Methodology:** Andrew Lauziere, Ryan Christensen, Hari Shroff, Radu Balan.

**Project administration:** Hari Shroff.

**Software:** Andrew Lauziere.

**Writing – original draft:** Andrew Lauziere.

**Writing – review & editing:** Andrew Lauziere, Ryan Christensen, Hari Shroff, Radu Balan.

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
