## [Decision Letter · Decision Letter 0]

29 Mar 2022

PONE-D-22-05770An Exact Hypergraph Matching Algorithm for Posture Identification in Embryonic C. elegansPLOS ONE

Dear Dr. Lauziere,

Thank you for submitting your manuscript to PLOS ONE. After careful consideration, we feel that it has merit but does not fully meet PLOS ONE’s publication criteria as it currently stands. Therefore, we invite you to submit a revised version of the manuscript that addresses the points raised during the review process.

We look forward to receiving your revised manuscript.

Kind regards,

Akbar Ali

Academic Editor

PLOS ONE

Journal Requirements:

2. Tables Not In Manuscript File (in the case of file type PDF)’ send back text: 'Please include your tables as part of your main manuscript and remove the individual files. Please note that supplementary tables (should remain/ be uploaded) as separate "supporting information" files

Reviewers' comments:

Reviewer's Responses to Questions

**Comments to the Author**

1. Is the manuscript technically sound, and do the data support the conclusions?

Reviewer #1: Yes

2. Has the statistical analysis been performed appropriately and rigorously? 

Reviewer #1: Yes

3. Have the authors made all data underlying the findings in their manuscript fully available?

Reviewer #1: Yes

4. Is the manuscript presented in an intelligible fashion and written in standard English?

Reviewer #1: Yes

5. Review Comments to the Author

Reviewer #1: In this manuscript, the authors proposed a new hypergraph-based matching algorithm for the posture identification of C. elegans. According to the authors, the proposed algorithm named EHGM can handle higher degree hypergraphs than existing point-set matching methods, leading to more accurate posture identification of C. elegans. The manuscript is well written, but for the layperson it is difficult to see the differences from existing methods, and the hypothesis and results seem unclear. The reviewer hopes that the comments below will contribute to improving the paper.

Major comments:

#1. The definition of “posture identification” is a bit unclear, making it difficult to interpret the results. Although the “top x accuracy” and “cost ratio” are explained in the text, it would be easier for the layperson to understand if they were shown in schema.

#2. Table 1, 2 and 3 compare the performance of 4 methods (KerGM, QAP, Pairs and Pusture), but the reviewer (and maybe the majority of readers) cannot intuitively recognize which method is EHGM. Therefore, a supplementary explanation would be helpful.

#3. The real microscopic image only appears in Figure 6. The reviewer believes that it would be very helpful to understand what each figure meant to say if the real pictures of nematodes were also displayed side-by-side in Figures 1 through 4.

#4. Figure 9, 10, 11 are not mentioned in the manuscript. Therefore, it is unclear in what context and for what purpose these figures are presented in the study.

Minor comments:

#1. Please indicate the full term of QAP (quadratic assignment problem?) when it first appears.

#2. In the legend of Fig. 10 and 11, some right parenthesis are missing.

#3. While the research paper is of a very high level of content, it is also notable for its redundancy. The Results and Methods sections describe what should be stated in the Introduction and Discussion sections, which undermines readability. The reviewer recommends to write only results in the Results section and only methods in the Methods section.

6. PLOS authors have the option to publish the peer review history of their article (what does this mean?). If published, this will include your full peer review and any attached files.

Reviewer #1: No

---

## [Author Response · Author response to Decision Letter 0]

3 May 2022

We have uploaded a rebuttal letter which contains our responses to both editor and reviewer comments.

---

## [Decision Letter · Decision Letter 1]

6 Sep 2022

PONE-D-22-05770R1An Exact Hypergraph Matching Algorithm for Posture Identification in Embryonic C. elegansPLOS ONE

Dear Dr. Lauziere,

Thank you for submitting your manuscript to PLOS ONE. After careful consideration, we feel that it has merit but does not fully meet PLOS ONE’s publication criteria as it currently stands. Therefore, we invite you to submit a revised version of the manuscript that addresses the points raised during the review process.

I would like to sincerely apologise for the delay you have incurred with your submission. It has been exceptionally difficult to secure reviewers to evaluate your study. We have now received two completed reviews; the comments are available below. Reviewer#1 has still significant scientific concerns about the study that need to be addressed in a revision.

Please revise the manuscript to address all the reviewer's comments in a point-by-point response in order to ensure it is meeting the journal's publication criteria. Please note that the revised manuscript will need to undergo further review, we thus cannot at this point anticipate the outcome of the evaluation process.

We look forward to receiving your revised manuscript.

Kind regards,

Miquel Vall-llosera Camps

Senior Editor

PLOS ONE

Additional Editor Comments:

PLOS ONE requires that submissions that present new methods as the primary focus of the manuscript must meet the criteria of utility, validation, and availability, which are described in detail at https://journals.plos.org/plosone/s/submission-guidelines#loc-methods-software-databases-and-tools. 

Reviewers' comments:

Reviewer's Responses to Questions

**Comments to the Author**

1. If the authors have adequately addressed your comments raised in a previous round of review and you feel that this manuscript is now acceptable for publication, you may indicate that here to bypass the “Comments to the Author” section, enter your conflict of interest statement in the “Confidential to Editor” section, and submit your "Accept" recommendation.

Reviewer #1: (No Response)

Reviewer #2: (No Response)

2. Is the manuscript technically sound, and do the data support the conclusions?

Reviewer #1: Yes

Reviewer #2: Yes

3. Has the statistical analysis been performed appropriately and rigorously? 

Reviewer #1: No

Reviewer #2: Yes

4. Have the authors made all data underlying the findings in their manuscript fully available?

Reviewer #1: Yes

Reviewer #2: Yes

5. Is the manuscript presented in an intelligible fashion and written in standard English?

Reviewer #1: Yes

Reviewer #2: Yes

6. Review Comments to the Author

Reviewer #1: In the revised manuscript, the authors have addressed some of the issues raised by the reviewer, but there is still much room for improvement in the way data are presented.

1. Does the title of Figure 1 correctly represent the content? It says that “low temporal resolution imaging necessitates posture identification”, but the reviewer thinks “high temporal resolution imaging rather necessitates posture identification”. “Failure of traditional tracking approaches” may better describe the content of the Figure 1.

2. Figure 3 is a visualization of Figures 1-B, so it is easier to understand when merged with Figure 1. Otherwise, it is recommended to rename it to Figure 2.

3. In all of Tables 1-3, it would be more formal and easier to understand if the rows and columns were interchanged. At first glance, it is difficult to understand what Top 1 to Top 10 represent, so a column of one higher level should be created to clearly indicate that they represent “Accuracy” (In the reviewer's personal opinion, either the Top 1 Accuracy or the Top 10 Accuracy should be sufficient to show in the Tables.).

4. It is not intuitively clear that the top chart and bottom chart of Table 2 correspond to the presence and absence of Q neuroblasts, respectively. The reviewer would like the authors to devise a better way to express this. Abbreviations for “Cost Ratio” and “Runtime” should be defined in legends for each table, or else they should be written in full terms.

5. The title of Table 3 says "seeding posterior pair identities reduce runtime", but does the Table really contain data on runtime?

6. Is the difference of accuracy between “Pairs” and “Posture” statistically significant?

Reviewer #2: The current version of the article is interesting and well-written. I only suggest to the authors putting convenient punctuation after each formula.

7. PLOS authors have the option to publish the peer review history of their article (what does this mean?). If published, this will include your full peer review and any attached files.

Reviewer #1: No

Reviewer #2: No

---

## [Author Response · Author response to Decision Letter 1]

27 Sep 2022

Our response has been uploaded "Response to Reviewers."

---

## [Decision Letter · Decision Letter 2]

17 Oct 2022

PONE-D-22-05770R2An Exact Hypergraph Matching Algorithm for Posture Identification in Embryonic C. elegansPLOS ONE

Dear Dr. Lauziere,

Thank you for your patience as this was reviewed, and for previously revising the manuscript in line with the reviewers' comments. After careful consideration, we feel that it has merit but requires one minor revision to meet PLOS ONE’s publication criteria as it currently stands. Therefore, we invite you to submit a revised version of the manuscript that addresses the points raised during the review process.

In short, after receiving the current round of reviews, we agreed with Comment #1 from Review 1 regarding the need to rework the abstract. The manuscript would be significantly improved if the motivating problem was made central to the abstract -- namely the difficulties and importance of estimating posture in C. elegans embryos. To increase the interest to the wider readership, this should precede the discussion of point matching. Once this has been addressed, I will be happy to fully recommend your paper for publication in PLOS ONE.

We look forward to receiving your revised manuscript.

Kind regards,

Barry L. Bentley, Ph.D.

Academic Editor

PLOS ONE

Journal Requirements:

Reviewers' comments:

Reviewer's Responses to Questions

**Comments to the Author**

1. If the authors have adequately addressed your comments raised in a previous round of review and you feel that this manuscript is now acceptable for publication, you may indicate that here to bypass the “Comments to the Author” section, enter your conflict of interest statement in the “Confidential to Editor” section, and submit your "Accept" recommendation.

Reviewer #1: All comments have been addressed

Reviewer #2: All comments have been addressed

2. Is the manuscript technically sound, and do the data support the conclusions?

Reviewer #1: Yes

Reviewer #2: Yes

3. Has the statistical analysis been performed appropriately and rigorously? 

Reviewer #1: Yes

Reviewer #2: Yes

4. Have the authors made all data underlying the findings in their manuscript fully available?

Reviewer #1: No

Reviewer #2: Yes

5. Is the manuscript presented in an intelligible fashion and written in standard English?

Reviewer #1: Yes

Reviewer #2: Yes

6. Review Comments to the Author

Reviewer #1: The authors appropriately addressed the concerns raised by the reviewers. The reviewer thinks the quality of the manuscript has now reached the acceptable level for publication. The reviewer would like to make few minor comments.

#1. Based on the update of the main manuscript, the reviewer thinks there is room to revise the ABSTRACT a little more. The current ABSTRACT begins with an explanation of point matching, but the reviewer thinks it would be easier for the reader to understand the flow of the discussion if the explanation began with the problems of posture estimation in C. elegans embryo. Also, how about adding some more concrete (or quantitative) experimental results?

#2. To ensure the reproducibility of the experimental results, might the authors consider making the datasets and programs used in this study publicly available? Are there any plans to implement the proposed method as a plug-in or add-in function for image analysis software such as ImageJ ?

Reviewer #2: (No Response)

7. PLOS authors have the option to publish the peer review history of their article (what does this mean?). If published, this will include your full peer review and any attached files.

Reviewer #1: No

Reviewer #2: No

---

## [Author Response · Author response to Decision Letter 2]

24 Oct 2022

We have attached a response to reviewers named "rebuttal_letter.docx."

---

## [Editor Report · Decision Letter 3]

26 Oct 2022

An Exact Hypergraph Matching Algorithm for Posture Identification in Embryonic C. elegans

PONE-D-22-05770R3

Dear Dr. Lauziere,

Thank you for submitting the latest corrections. We’re pleased to inform you that your manuscript has been judged scientifically suitable for publication and will be formally accepted for publication.

Within one week, you’ll receive an e-mail detailing any final required amendments. When these have been addressed, you’ll receive a formal acceptance letter and your manuscript will be scheduled for publication.

Kind regards,

Barry L. Bentley, Ph.D.

Academic Editor

PLOS ONE
---

## [Editor Report · Acceptance letter]

2 Nov 2022

PONE-D-22-05770R3 

An Exact Hypergraph Matching Algorithm for Posture Identification in Embryonic *C. elegans*

Dear Dr. Lauziere:

I'm pleased to inform you that your manuscript has been deemed suitable for publication in PLOS ONE. Congratulations! Your manuscript is now with our production department. 

Kind regards, 

on behalf of

Dr Barry L. Bentley 

Academic Editor

PLOS ONE